# An SU(2) Gauge Principle for the Cosmic Microwave Background: Perspectives on the Dark Sector of the Cosmological Model

**Ralf Hofmann**

Institut für Theoretische Physik, Universität Heidelberg, Philosophenweg 16, 69120 Heidelberg, Germany;
r.hofmann@thphys.uni-heidelberg.de

**Abstract:** We review consequences for the radiation and dark sectors of the cosmological model arising from the postulate that the Cosmic Microwave Background (CMB) is governed by an SU(2) rather than a U(1) gauge principle. We also speculate on the possibility of actively assisted structure formation due to the de-percolation of lump-like configurations of condensed ultralight axions with a Peccei–Quinn scale comparable to the Planck mass. The chiral-anomaly induced potential of the axion condensate receives contributions from SU(2)/SU(3) Yang–Mills factors of hierarchically separated scales which act in a screened (reduced) way in confining phases.

**Keywords:** light scalar fields; axial anomaly; SU(2) Yang–Mills thermodynamics; de-percolation of axionic lumps; cosmological and galactic dark-matter densities

## 1. Introduction

Judged by decently accurate agreement of cosmological parameter values extracted from (i) large-scale structure observing campaigns on galaxy- and shear-correlation functions as well as redshift-space distortions towards a redshift of unity by photometric/spectroscopic surveys (sensitive to Baryonic Accoustic Oscillations (BAO), the evolution of Dark Energy, and non-linear structure growth), for recent, ongoing, and future projects see, e.g., [1–5]; (ii) fits of various CMB angular power spectra based on frequency-band optimised intensity and polarisation data collected by satellite missions [6–8]; and (iii) fits to cosmologically local luminosity-distance redshift data for Supernovae Ia (SNeIa) [9,10] the spatially flat standard Lambda Cold Dark Matter (ΛCDM) cosmological model is a good, robust starting point to address the evolution of our Universe. About twenty years ago, the success of this model has triggered a change in paradigm in accepting a present state of accelerated expansion induced by an essentially dark Universe made of 70% Dark Energy and 25% Dark Matter.

While (i) and (ii) are anchored on comparably large cosmological standard-ruler co-moving distance scales, the sound horizons $r_{s,d}$ and $r_{s,*}$, which emerge at baryon-velocity freeze-out and recombination, respectively, during the epoch of CMB decoupling and therefore refer to very-high-redshift physics ($z > 1000$), (iii) is based on direct distance ladders linking to host galaxies at redshifts of say, $0.01 < z < 2.3$, which do not significantly depend on the assumed model for Dark Energy [11] and are robust against sample variance, local matter-density fluctuations, and directional bias [12–15].

Thanks to growing data quality and increased data-analysis sophistication to identify SNeIa [16] and SNeII [17] spectroscopically, to establish precise and independent geometric distance indicators [18,19] (e.g., Milky Way parallaxes, Large Magellanic Cloud (LMC) detached eclipsing binaries, and masers in NGC 4258), tightly calibrated period-to-luminosity relations for LMC cepheids and cepheids in SNeIa host galaxies [19], the use of the Tip of the Red Giant Branch (TRGB) [20,21] or the

Asymptotic Giant Branch (AGB, Mira) [22] to connect to the distance ladder independently of cepheids, the high value of the basic cosmological parameter $H_0$ (Hubble expansion rate today) has (see [21], however), over the last decade and within $\Lambda$CDM, developed a tension of up to $\sim 4.4\,\sigma$ [19] in SNeIa distance-redshift fits using the LMC geometrically calibrated cepheid distance ladder compared to its low value extracted from statistically and systematically accurate medium-to-high-l CMB angular power spectra [7,8] and the BAO method [3]. Both, the CMB and BAO probe the evolution of small radiation and matter density fluctuations in global cosmology: Fluctuations are triggered by an initial, very-high-redshift (primordial) spectrum of scalar/tensor curvature perturbations which, upon horizon entry, evolve linearly [23] up to late times when structure formation generates non-linear contributions in the galaxy spectra [2,3] and subjects the CMB to gravitational lensing [24].

Recently, a cosmographic way of extracting $H_0$ through time delays of strongly lensed high-redshift quasars, see e.g., [25,26], almost matches the precision of the presently most accurate SNeIa distance-redshift fits [19]—$H_0 = 71.9^{+2.4}_{-3.0}$ km s$^{-1}$ Mpc$^{-1}$ vs. $H_0 = (74.22 \pm 1.82)$ km s$^{-1}$ Mpc$^{-1}$—supporting a high local value of the Hubble constant and rendering the local-global tension even more significant [27]. The high local value of $H_0 = (72 \cdots 74)$ km s$^{-1}$ Mpc$^{-1}$ [19,27] (compared to the global value of $H_0 = (67.4 \pm 0.5)$ km s$^{-1}$ Mpc$^{-1}$ [8]) is stable to sources of systematic uncertainty such as line-of-sight effects, peculiar motion (stellar kinematics), and assumptions made in the lens model. There is good reason to expect that an improved localisation of sources for gravitational-wave emission without an electromagnetic counterpart and the increase of statistics in gravitational-wave events accompanied by photon bursts (standard sirens) within specific host galaxies will lead to luminosity-distance-high-redshift data producing errors in $H_0$ comparable of those of [19], independently of any distance-ladder calibration [28], see also [29].

Apart from the Hubble tension, there are smaller local-global but possibly also tensions between BAO and the CMB in other cosmological parameters such as the amplitude of the density fluctuation power spectrum ($\sigma_8$) and the matter content ($\Omega_m$), see [30]. Finally, there are persistent large-angle anomalies in the CMB, already seen by the Cosmic Background Explorer (COBE) and strengthened by the Wilkinson Microwave Anisotropy Probe (WMAP) and Planck satellite, whose atoms (a) lack of correlation in the TT two-point function, (b) a rather significant alignment of the low multipoles (*p*-values of below 0.1%), and (c) a dipolar modulation, which is independent of the multipole alignment (b), indicate a breaking of statistical isotropy at low angular resolution, see [31] for a comprehensive and complete review.

The present work intends to review and discuss a theoretical framework addressing a possibility to resolve the above-sketched situation. The starting point is to subject the CMB to the thermodynamics of an extended gauge principle in replacing the conventional group U(1) by SU(2). Motivated [32] by explaining an excess in CMB line temperature at radio frequencies, see [33] and references therein, this postulate implies a modified temperature-redshift relation which places CMB recombination at a redshift of $\sim$1700 rather than $\sim$1100 and therefore significantly reduces the density of matter at that epoch. To match $\Lambda$CDM at low $z$, one requires a release of Dark Matter from early Dark Energy within the dark ages at a redshift $z_p$: lumps and vortices, formely tighly correlated within a condensate of ultralight axion particles, de-percolate into independent pressureless (and selfgravitating) solitons due to cosmological expansion, thereby contributing actively to non-linear structure formation at low $z$. A fit to CMB angular power spectra of a cosmological model, which incorporates these SU(2) features on the perfect-fluid level but neglects low-$z$ radiative effects in SU(2) [34], over-estimates TT power on large angular scales but generates a precise fit to the data for $l \geq 30$. With $z_p \sim 53$ a locally favoured value of $H_0 \sim 74$ km s$^{-1}$ Mpc$^{-1}$ and a low baryon density $\omega_{b,0} \sim 0.017$ are obtained. Moreover, the conventionally extracted near scale invariance of adiabatic, scalar curvature perturbations comes out to be significantly broken by an infrared enhancement of their power spectrum. Finally, the fitted redshift $\sim$6 of re-ionisation is compatible with the detection of the Gunn-Peterson-trough in the spectra of high-$z$ quasars [35] and distinctly differs from the high value of CMB fits to $\Lambda$CDM [7,8].

As of yet, there are loose ends to the SU(2) based scenario. Namely, the physics of de-percolation requires extra initial conditions for matter density fluctuations at $z_p$. In the absence of a precise modelling of the 'microscopics' of the associated soliton ensembles[1] it is only a guess that these fluctuations instantaneously follow the density fluctuations of primordial Dark Matter as assumed in [34]. Moreover, it is necessary to investigate to what extent profiles of the axion field (lumps of localised energy density) actively seed non-linear structure formation and whether their role in galactic halo formation and baryon accretion meets the observational constraints (Tully–Fisher relation, etc.). Furthermore, radiative effects on thermal photon propagation at low $z$ [36], which are expected to contribute or even explain the above-mentioned atoms (a), (b), and (c) of the CMB large-angle anomalies, see [37], and which could reduce the excess in low-$l$ TT power of [34] to realistic levels, need to be incorporated into the model. They require analyses in terms of Maxwell's multipole-vector formalism [31] and/or other robust and intuitive statistics to characterise multipole alignment and the large-angle suppression of TT.

This review-type paper is organised as follows. In Section 2 we briefly discuss the main distinguishing features between the conventional ΛCDM model and cosmology based on a CMB which obeys deconfining SU(2) Yang–Mills thermodynamics. A presentation of our recent results on angular-power-spectra fits to Planck data is carried out Section 3, including a discussion of the parameters $H_0$, $n_s$, $\sigma_8$, $z_{re}$, and $\omega_{b,0}$. In Section 4 we interpret the rough characteristics of the Dark Sector employed in [34] to match ΛCDM at low $z$. In particular, we point out that the value of $z_p$ seems to be consistent with the typical dark-matter densities in the Milky Way. Finally, in Section 5 we sketch what needs to be done to arrive at a solid, observationally well backed-up judgement of whether SU(2)$_{CMB}$ based cosmology (and its extension to SU(2) and SU(3) factors of hierarchically larger Yang–Mills scales including their nonthermal phase transitions) may provide a future paradigm to connect local cosmology with the very early Universe. Throughout the article, super-natural units $\hbar = k_B = c = 1$ are used.

## 2. SU(2)$_{CMB}$ vs. Conventional CMB Photon Gas in ΛCDM

The introduction of an SU(2) gauge principle for the description of the CMB is motivated theoretically by the fact that the deconfining thermodynamics of such a Yang–Mills theory exhibits a thermal ground state, composed of densely packed (anti)caloron [38] centers with overlapping peripheries [39,40], which breaks SU(2) to U(1) in terms of an adjoint Higgs mechanism [41]. Therefore, the spectrum of excitations consists of one massless gauge mode, which can be identified with the CMB photon, and two massive vector modes of a temperature-dependent mass on tree level: thermal quasi-particle excitations. The interaction between these excitations is feeble [41]. This is exemplified by the one-loop polarisation tensor of the massless mode [36,42,43]. As a function of temperature, polarisation effects peak at about twice the critical temperature $T_c$ for the deconfining-preconfining transition. As a function of increasing photon momentum, there are regimes of radiative screening/ antiscreening, the latter being subject to an exponential fall-off [36]. At the phase boundary ($T \sim T_c$) electric monopoles [41], which occur as isolated and unresolved defects deeply in the deconfining phase, become massless by virtue of screening due to transient dipoles [44] and therefore condense to endow the formely massless gauge mode with a quasiparticle Meissner mass $m_\gamma$. This mass rises critically (with mean-field exponent) as $T$ falls below $T_c$ [41]. Both, (i) radiative screening/antiscreening of massless modes and (ii) their Meissner effect are important handles in linking SU(2)$_{CMB}$ to the CMB: while (i) induces large-ange anomalies into the TT correlation [37] and contributes dynamically to the CMB dipole [45,46] (ii) gives rise to a nonthermal spectrum

---

[1]　Lump sizes could well match those of galactic dark-matter halos, see Section 4.

of evanescent modes[2] for frequencies $\omega < m_\gamma$ once $T$ falls below $T_c$. This theoretical anomaly of the blackbody spectrum in the Rayleigh–Jeans regime can be considered to explain the excess in CMB radio power below 1 GHz, see [33] and references therein, thereby fixing $T_c = 2.725$ K and, as a consequence of $\lambda_c = 13.87 = \frac{2\pi T_c}{\Lambda_{\text{CMB}}}$ [41], the Yang–Mills scale of SU(2)$_{\text{CMB}}$ to $\Lambda_{\text{CMB}} \sim 10^{-4}$ eV [32].

Having discussed the low-frequency deviations of SU(2)$_{\text{CMB}}$ from the conventional Rayleigh–Jeans spectrum, which fix the Yang–Mills scale and associate with large-angle anomalies, we would now like to review its implications for the cosmological model. Of paramount importance for the set-up of such a model is the observtion that SU(2)$_{\text{CMB}}$ implies a modified temperature ($T$)-redshift ($z$) relation for the CMB which is derived from energy conservation of the SU(2)$_{\text{CMB}}$ fluid in the deconfining phase in an FLRW universe with scale factor $a$ normalised to unity today [47]. Denoting by $T_c = T_0 = 2.725$ K [32] the present CMB baseline temperature [6] and by $\rho_{\text{SU(2)}_{\text{CMB}}}$ and $P_{\text{SU(2)}_{\text{CMB}}}$ energy density and pressure, respectively, of SU(2)$_{\text{CMB}}$, one has

$$a \equiv \frac{1}{z+1} = \exp\left(-\frac{1}{3}\log\left(\frac{s_{\text{SU(2)}_{\text{CMB}}}(T)}{s_{\text{SU(2)}_{\text{CMB}}}(T_0)}\right)\right), \tag{1}$$

where the entropy density $s_{\text{SU(2)}_{\text{CMB}}}$ is defined by

$$s_{\text{SU(2)}_{\text{CMB}}} \equiv \frac{\rho_{\text{SU(2)}_{\text{CMB}}} + P_{\text{SU(2)}_{\text{CMB}}}}{T}. \tag{2}$$

For $T \gg T_0$, Equation (1) simplifies to

$$T = \left(\frac{1}{4}\right)^{1/3} T_0(z+1) \approx 0.63\, T_0(z+1). \tag{3}$$

For arbitrary $T \geq T_0$, a multiplicative deviation $S(z)$ from linear scaling in $z+1$ can be introduced as

$$S(z) = \left(\frac{\rho_{\text{SU(2)}_{\text{CMB}}}(z=0) + P_{\text{SU(2)}_{\text{CMB}}}(z=0)}{\rho_{\text{SU(2)}_{\text{CMB}}}(z) + P_{\text{SU(2)}_{\text{CMB}}}(z)}\frac{T^4(z)}{T_0^4}\right)^{1/3}. \tag{4}$$

Therefore,

$$T = S(z)\, T_0(z+1). \tag{5}$$

Figure 1 depicts function $S(z)$.

Amusingly, the asymptotic $T$-$z$ relation of Equation (3) also holds for the relation between $\rho_{\text{SU(2)}_{\text{CMB}}}(z)$ and the conventional CMB energy density $\rho_\gamma(z)$ in $\Lambda$CDM (the energy density of a thermal U(1) photon gas, using the $T$-$z$ relation $T = T_0(z+1)$). Namely,

$$\rho_{\text{SU(2)}_{\text{CMB}}}(z) = 4\left(\frac{1}{4}\right)^{4/3}\rho_\gamma(z) = \left(\frac{1}{4}\right)^{1/3}\rho_\gamma(z) \quad (z \gg 1). \tag{6}$$

Therefore, the (gravitating) energy density of the CMB in SU(2)$_{\text{CMB}}$ is, at the same redshift $z \gg 1$, by a factor of $\sim 0.63$ smaller than that of the $\Lambda$CDM model even though there are eight (two plus two times three) gauge-mode polarisations in SU(2)$_{\text{CMB}}$ and only two such polarisations in the U(1) photon gas.

---

[2] That the deep Rayleigh–Jeans regime is indeed subject to classical wave propagation is assured by the fact that wavelengths that are greater than the spatial scale $s \equiv \pi T |\phi|^{-2}$, separating a(n) (anti)caloron center from its periphery where its (anti)selfdual gauge field is that of a dipole [40]. The expression for $s$ contains the modulus $|\phi| = \sqrt{\Lambda_{\text{CMB}}^3/(2\pi T)}$ of the emergent, adjoint Higgs field $\phi$ ($\Lambda_{\text{CMB}} \sim 10^{-4}$ eV the Yang–Mills scale of SU(2)$_{\text{CMB}}$), associated with densely packed (anti)caloron centers, and, explicitely, temperature $T$.

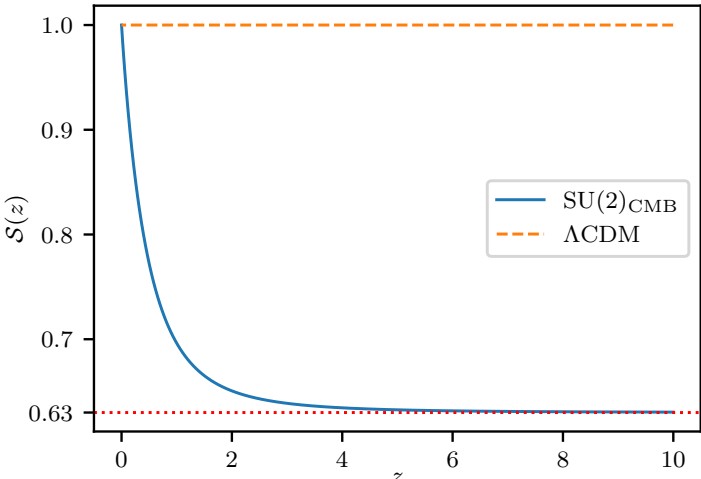

**Figure 1.** The function $\mathcal{S}(z)$ of Equation (4), indicating the (multiplicative) deviation from the asymptotic $T$-$z$ relation in Equation (3). Curvature in $\mathcal{S}(z)$ for low $z$ arises from a breaking of scale invariance for $T \sim T_0 = T_c$. There is a rapid approach towards the asymptotics $\left(\frac{1}{4}\right)^{1/3} \approx 0.63$ with increasing $z$. Figure adopted from [34].

Not yet considering linear and next-to-linear perturbations in $\mathrm{SU(2)_{CMB}}$ to shape typical CMB large-angle anomalies in terms of late-time screening/antiscreening effects [37], Equation (5) has implications for the Dark Sector if one wishes to maintain the successes of the standard cosmological model at low $z$ where local cosmography and fits to distance-redshift curves produce a consistent framework within $\Lambda$CDM. As it was shown in [34], the assumption of matter domination at recombination, which is not unrealistic even for $\mathrm{SU(2)_{CMB}}$ [34], implies that

$$z_{\Lambda\mathrm{CDM},*} \sim \left(\frac{1}{4}\right)^{1/3} z_{\mathrm{SU(2)_{CMB}},*} \tag{7}$$

and, as a consequence,

$$\Omega_{\Lambda\mathrm{CDM},m,0} \approx 4\,\Omega_{\mathrm{SU(2)_{CMB}},m,0}\,, \tag{8}$$

where $\Omega_{m,0}$ denotes today's density parameter for nonrelativistic matter (Dark Matter plus baryons), and $z_*$ is the redshift of CMB photon decoupling in either model. Since the matter sector of the $\mathrm{SU(2)_{CMB}}$ model, as roughly represented by Equation (8), contradicts $\Lambda$CDM at low $z$ one needs to allow for a transition between the two somewhere in the dark ages as $z$ decreases. In [34] a simple model, where the transition is sudden at a redshift $z_p$ and maintains a small dark-energy residual, was introduced: a coherent axion field—a dark-energy like condensate of ultralight axion particles, whose masses derive from $\mathrm{U(1)}_A$ anomalies [48–52] invoked by the topological charges of (anti)caloron centers in the thermal ground state of $\mathrm{SU(2)_{CMB}}$ and, in a screened way, $\mathrm{SU(2)/SU(3)}$ Yang–Mills factors of higher scales—releases solitonic lumps by de-percolation due to the Universe's expansion. Accordingly, we have

$$\Omega_{\mathrm{ds}}(z) = \Omega_\Lambda + \Omega_{\mathrm{pdm},0}(z+1)^3 + \Omega_{\mathrm{edm},0} \begin{cases} (z+1)^3, & z < z_p \\ (z_p+1)^3, & z \geq z_p \end{cases}. \tag{9}$$

Here $\Omega_\Lambda$ and $\Omega_{\mathrm{pdm},0} + \Omega_{\mathrm{edm},0} \equiv \Omega_{\mathrm{cdm},0}$ represent today's density parameters for Dark Energy and Dark Matter, respectively, $\Omega_{\mathrm{pdm},0}$ refers to primordial Dark Matter (that is, dark matter that existed *before* the initial redshift of $z_i \sim 10^4$ used in the CMB Boltzmann code) for all $z$ and $\Omega_{\mathrm{edm},0}$ to emergent Dark Matter for $z < z_p$. In [34] the initial conditions for the evolution of density and velocity

perturbations of the emergent Dark-Matter portion at $z_p$ are, up to a rescaling of order unity, assumed to follow those of the primordial Dark Matter.

### 3. SU(2)$_{\text{CMB}}$ Fit of Cosmological Parameters to Planck Data

In [34] a simulation of the new cosmological model subject to SU(2)$_{\text{CMB}}$ and the Dark Sector of Equation (9) was performed using a modified version of the Cosmic Linear Anisotropy Solving System (CLASS) Boltzmann code [53]. Best fits to the 2015 Planck data [8] on the angular power spectra of the two-point correlation functions temperature–temperature (TT), electric-mode polarisation–electric-mode polarisation (EE), and temperature–electric-mode polarisation (TE), subject to typical likelihood functions used by the Planck collaboration, were performed. Because temperature perturbations can only be coherently propagated by (low-frequency) massless modes in SU(2)$_{\text{CMB}}$ [40] the propagation of the (massive) vector modes was excluded in one version of the code (physically favoured). Furthermore, entropy conservation in $e^+e^-$ annihilation invokes a slightly different counting of relativistic degrees of freedom in SU(2)$_{\text{CMB}}$ for this epoch [54]. As a consequence, we have a $z$-dependent density parameter for (massless) neutrinos given as [34]

$$\Omega_\nu(z) = \frac{7}{8} N_{\text{eff}} \left(\frac{16}{23}\right)^{\frac{4}{3}} \Omega_{\text{SU(2)}_{\text{CMB}},\gamma}(z)\,, \tag{10}$$

where $N_{\text{eff}}$ refers to the effective number of neutrino flavours (or any other extra relativistic, free-streaming, fermionic species), and $\Omega_{\text{SU(2)}_{\text{CMB}},\gamma}$ is the density parameter associated with the massless mode only in the expression for $\rho_{\text{SU(2)}_{\text{CMB}}}(z)$ of Equation (6). The value $N_{\text{eff}} = 3.046$ of the Planck result was used as a fixed input in [34]. The statistical goodness of the best fit was found to be comparable to the one obtained by the Planck collaboration [8], see Figure 2 as well as the lower part of Table 1. There is an excess of power in TT for $7 \leq l \leq 30$, however.

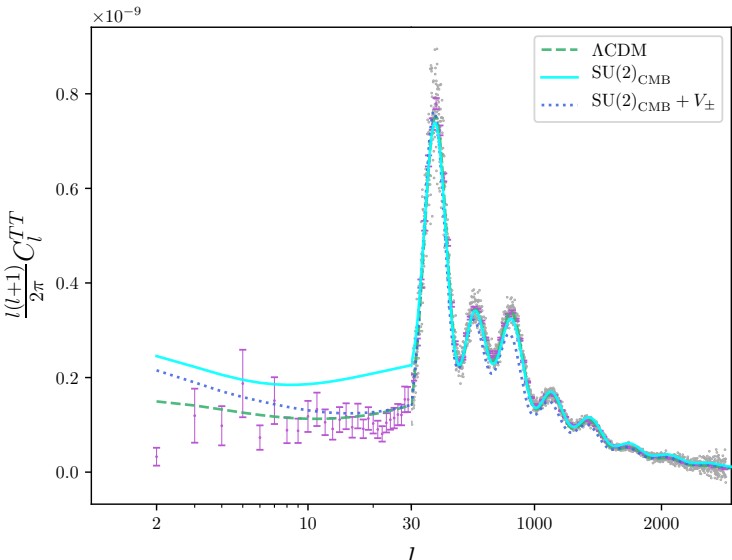

**Figure 2.** Normalised power spectra of TT correlator for best-fit parameter values quoted in Table 1: Dashed, dotted, and solid lines represent $\Lambda$CDM, SU(2)$_{\text{CMB}} + V_\pm$ (not considered in in the present work), and SU(2)$_{\text{CMB}}$, respectively. For $l \leq 29$ the 2015 Planck data points are unbinned and carry error bars, for $l \geq 30$ grey points represent unbinned spectral power. Figure adopted from [34].

This excess could be attributable to the omission of radiative effects in the low-$z$ propagation of the massless mode. The consideration of the modified dispersion law into the CMB Boltzmann code presently is under way. The following table was obtained in [34]:

**Table 1.** Best-fit cosmological parameters of the $SU(2)_{CMB}$ and the $\Lambda$CDM model as obtained in [34]. The best-fit parameters of $\Lambda$CDM together with their 68% confidence intervals are taken from [8], employing the TT,TE,EE+lowP+lensing likelihoods. For $SU(2)_{CMB}$ the HiLLiPOP+lowTEB+lensing likelihood (lowP and lowTEB are pixel-based likelihoods) was used, see [55]. The central values associate with $\chi^2 = \chi^2_{ll} + \chi^2_{hl}$ (best fit) as quoted in the lower part of the Table.

| Parameter | $SU(2)_{CMB}$ | $\Lambda$CDM |
|---|---|---|
| $\omega_{b,0}$ | $0.0173 \pm 0.0002$ | $0.0225 \pm 0.00016$ |
| $\omega_{pdm,0}$ | $0.113 \pm 0.002$ | – |
| $\omega_{edm,0}$ | $0.0771 \pm 0.0012$ | – |
| $100\,\theta_*$ | $1.0418 \pm 0.0022$ | $1.0408 \pm 0.00032$ |
| $\tau_{re}$ | $0.02632 \pm 0.00218$ | $0.079 \pm 0.017$ |
| $\ln(10^{10} A_s)$ | $2.858 \pm 0.009$ | $3.094 \pm 0.034$ |
| $n_s$ | $0.7261 \pm 0.0058$ | $0.9645 \pm 0.0049$ |
| $z_p$ | $52.88 \pm 4.06$ | – |
| $\beta$ | $0.811 \pm 0.058$ | – |
| $H_0 / _{km\,s^{-1}Mpc^{-1}}$ | $74.24 \pm 1.46$ | $67.27 \pm 0.66$ |
| $z_{re}$ | $6.23^{+0.41}_{-0.42}$ | $10^{+1.7}_{-1.5}$ |
| $z_*$ | $1715.19 \pm 0.19$ | $1090.06 \pm 0.30$ |
| $z_d$ | $1640.87 \pm 0.27$ | $1059.65 \pm 0.31$ |
| $\omega_{cdm,0}$ | $0.1901 \pm 0.0023$ | $0.1198 \pm 0.0015$ |
| $\Omega_\Lambda$ | $0.616 \pm 0.006$ | $0.6844 \pm 0.0091$ |
| $\Omega_{m,0}$ | $0.384 \pm 0.006$ | $0.3156 \pm 0.0091$ |
| $\sigma_8$ | $0.709 \pm 0.020$ | $0.8150 \pm 0.0087$ |
| Age/Gyr | $11.91 \pm 0.10$ | $13.799 \pm 0.021$ |
| $\chi^2_{ll}$ | 10,640 | 10,495 |
| $n_{dof,ll}$ | 9207 | 9210 |
| $\frac{\chi^2_{ll}}{n_{dof,ll}}$ | 1.156 | 1.140 |
| $\chi^2_{hl}$ | 10,552.6 | 9951.47 |
| $n_{dof,hl}$ | 9547 | 9550 |
| $\frac{\chi^2_{hl}}{n_{dof,hl}}$ | 1.105 | 1.042 |

We would like to discuss the following parameters (the value of $z_p$ will be discussed in Section 4): (i) $H_0$, (ii) $n_s$, (iii) $\sigma_8$, (iv) $z_{re}$, and (v) $\omega_{b,0}$. (i): The fitted value of $H_0$ is in agreement with local-cosmology observation [19,27] but discrepant at the $4.5\,\sigma$ level with the value $H_0 \sim (67.4 \pm 0.5)\,km\,Mpc^{-1}\,s^{-1}$ in $\Lambda$CDM of global-cosmology fits as extracted by the Planck collaboration [8], a galaxy-custering survey—the Baryonic Oscillation Spectroscopy Survey (BOSS) [3]—and a galaxy-custering-weak-lensing survey—the Dark Energy Survey (DES) [2]—whose distances are derived from inverse distance ladders using the sound horizons at CMB decoupling or baryon drag as references. Such anchors assume the validity of $\Lambda$CDM (or a variant thereof with variable Dark Energy) at high $z$, and the data analysis employs fiducial cosmologies that are close to those of CMB fits. As a rule of thumb, the inclusion of extra relativistic degrees of freedom within reasonable bounds [2,3] but also sample variance, local matter-density fluctuations, or a directional bias in SNa Ia observations [12–15] cannot explain the above-quoted tension in $H_0$. Note that the DES Y1 data on three two-point functions (cosmic shear auto correlation, galaxy angular auto correlation, and galaxy-shear cross correlation in up to five redshift bins with $z \leq 1.3$), which are most sensitive to $\Omega_m$ and $\sigma_8$ (or $S_8 = \sigma_8/\Omega_m$), cannot usefully constrain $H_0$ by itself. However, the combination of DES 1Y data with those of Planck (no lensing) yields an increases in $H_0$ on the 1-2$\,\sigma$ level compared to the Planck (no lensing) data alone, and a similar tendency is seen when BAO data [3] are enriched with those of DES Y1, see Table II in [2]. Loosely speaking, one thus may state a mild increase of $H_0$ with an

increasing portion of late-time (local-cosmology) information in the data. (ii): The index of the initial spectrum of adiabatic, scalar curvature perturbations $n_s$ is unusually low, expressing an enhancement of infrared modes as compared to the ultraviolet ones (violation of scale invariance). As discussed in [34], such a tilted spectrum is not consistent with single-field slow-roll inflation and implies that the Hubble parameter during the inflationary epoch has changed appreciably. (iii): There is a low value of $\sigma_8$ (initial amplitude of matter-density power spectrum at comoving wavelength $8\,h^{-1}\,$Mpc with $H_0 \equiv h\,100\,\mathrm{km\,s^{-1}Mpc^{-1}}$) compared to CMB fits [8] and BAO [3] although the DES Y1 fit to cosmic shear data alone would allow for a value $\sigma_8 \sim 0.71$ within the 1-$\sigma$ margin [2]. This goes also for our high value of today's matter density parameter $\Omega_{m,0}$ (the ratio of physical matter density to the critical density). (iv): The low value of the redshift to (instantaneous) re-ionisation $z_{re}$ (and the correspondingly low optical depth $\tau_{re}$) compared to values obtained in CMB fits [8] is consistent with the one extracted from the observation of the Gunn-Peterson trough in the spectra of high-redshift quasars [35]. (v): The low value of $\omega_{b,0}$—today's physical baryon density—could provide a theoretical solution of the missing baryon problem [34] (missing compared to CMB fits to $\Lambda$CDM (CMB fits) and the primordial D/H (Deuterium-to-Hydrogen) ratio in Big-Bang Nucleosynthesis (BBN), the latter yielding $\omega_{b,0} = 0.02156 \pm 0.00020$ [56] which is by $2.3\,\sigma$ lower than the Planck value of $0.0225 \pm 0.00016$ in Table 1). If this low value of $\omega_{b,0}$ could be consolidated observationally than either the idea that BBN is determined by isolated nuclear reaction cross sections only or the observation of a truly primordial D/H in metal-poor environments or both will have to be questioned in future ivestigations. There is a recent claim, however, that the missing 30–40% of baryons were observed in highly ionized oxygen absorbers representing the warm-hot intergalactic medium when illuminated by the X-ray spectrum of a quasar with $z > 0.4$ [57]. This results is disputed in [58]. In [59] the missing baryon problem is addressed by the measurement of electron column density within the intergalactic medium using the dispersion of localised radio bursts with $z \le 0.522$. In projecting electron density the measurement appeals to a flat $\Lambda$CDM CMB-fitted model (Planck collaboration [8]). Presently, $\omega_{b,0}$ is extracted from the ASKAP data in [59] to be consistent with CMB fits and BBN albeit subject to a 50% error which is expected to decrease with the advent of more powerful radio observatories such as SKA.

## 4. Axionic Dark Sector and Galactic Dark-Matter Densities

The model for the Dark Sector in Equation (9) is motivated by the possibility that a coherent condensate of ultra-light particles—an axion field [52]—forms selfgravitating lumps or vortices in the course of nonthermal (Hagedorn) phase transitions due to SU(2)/SU(3) Yang–Mills factors, governing the radiation and matter content of the very early Universe, going confining. A portion of the thus created, large abundance of such solitons percolates into a dark-energy like state: the contributions $\Omega_\Lambda$ and $\Omega_{edm,0}(z_p + 1)^3$ in Equation (9) of which the latter may de-percolate at $z = z_p$ into a dark-matter like component $\Omega_{edm,0}(z + 1)^3$ for $z < z_p$, the expansion of the Universe increasing the average distance between the centers of neighbouring solitons. By virtue of the $U(1)_A$ anomaly, mediated by topological charge density [48–51], residing in turn within the ground state of an SU(2)/SU(3) Yang–Mills theory [41], the mass $m_a$ of an axion particle due to a single such theory of Yang–Mills scale $\Lambda$ is given as [52]

$$m_a = \frac{\Lambda^2}{M_P}, \tag{11}$$

where we have assumed that the so-called Peccei–Quinn scale, which associates with a dynamical chiral-symmetry breaking, was set equal to the Planck mass $M_P = 1.221 \times 10^{28}\,$eV, see [60,61] for motivations. Let (natural units: $c = \hbar = 1$)

$$r_c \equiv 1/m_a \tag{12}$$

denote the Compton wavelength,

$$r_B \equiv \frac{M_P^2}{M} m_a^{-2} \tag{13}$$

the gravitational Bohr radius, where $M \sim 10^{12}\,M_\odot$ is the total dark mass of a typical (spiral) galaxy like the Milky Way, and

$$d_a \equiv \left( \frac{m_a}{\rho_{\rm dm}} \right)^{1/3} \tag{14}$$

the interparticle distance where $\rho_{\rm dm}$ indicates the typical mean energy density in Dark Matter of a spiral. Following [62], we assume

$$\rho_{\rm dm} = (0.2 \cdots 0.4)\,{\rm GeV\,cm^{-3}}. \tag{15}$$

For the concept of a gravitational Bohr radius in a selfconsistent, non-relativistic potential model to apply, the axion-particle velocity $v_a$ in the condensate should be much small than unity. Appealing to the virial theorem at a distance to the gravitational center of $r_B$, one has $v_a \sim \left( \frac{Mm_a}{M_P^2} \right)^2$ [63]. In [63], where a selfgravitating axion condensate was treated non-relativistically by means of a non-linear and non-local Schrödinger equation to represent a typical galactic dark-matter halo, the following citeria on the validity of such an approach were put forward (natural units: $c = \hbar = 1$): (i) $v_a \ll 1$. (ii) $d_a \ll r_c$ is required for the description of axion particles in terms of a coherent Bose condensate to be realistic. (iii) $r_B$ should be the typical extent of a galactic Dark-Matter halo: $r_B \sim 100$–300 kpc. With $\Lambda_{\rm CMB} \sim 10^{-4}\,{\rm eV}$ one obtains $m_a = 8.2 \times 10^{-37}\,{\rm eV}$, $d_a = (4.1 \cdots 5.2) \times 10^{-34}\,{\rm pc}$, $r_c = 3.2 \times 10^{25}\,{\rm pc}$, and $r_B \sim 8 \times 10^{24}\,{\rm kpc}$. While (i) and (ii) are extremely well satified with $v_a \sim 10^{-28}$ and $\frac{d_a}{r_c} \sim 10^{-59}$ point (iii) is badly violated. ($r_B$ is about $2 \times 10^{18}$ the size of the visible Universe).

Therefore, the Yang–Mills scale responsible for the axion mass that associates with dark-matter halos of galaxies must be dramatically larger. Indeed, setting $\Lambda = 10^{-2}\Lambda_e$ where $\Lambda_e = \frac{m_e}{118.6}$ is the Yang–Mills scale of an SU(2) theory that could associate with the emergence of the electron of mass $m_e = 511\,{\rm keV}$ [64,65], one obtains $m_a = 1.5 \times 10^{-25}\,{\rm eV}$, $d_a = (2.3 \cdots 2.9) \times 10^{-30}\,{\rm pc}$, $r_c = 1.7 \times 10^{14}\,{\rm pc}$, and $r_B \sim 232\,{\rm kpc}$. In addition to (i) and (ii) with $v_a \sim 10^{-4}$ and $\frac{d_a}{r_c} \sim 10^{-44}$ also point (iii) is now well satisfied. If the explicit Yang–Mills scale of an SU(2) theory, which is directly imprinted in the spectra of the excitations in the pre - and deconfining phases, acts only in a screened way in the confining phase as far as the axial anomaly is concerned—reducing its value by a factor of one hundred or so—then the above axionic Dark-Sector scenario would link the theory responsible for the emergence of the electron with galactic dark-matter halos! In addition, the axions of SU(2)$_{\rm CMB}$ would provide the Dark-Energy density $\Omega_\Lambda$ of such a scenario.

Finally, we wish to point out that the de-percolation mechanism of axionic solitons (lumps forming out of former Dark-Energy density) in the cosmological model based on SU(2)$_{\rm CMB}$, which may be considered to underly the transition in the Dark Sector at $z_p = 53$ described by Equation (9), is consistent with the Dark-Matter density in the Milky Way. Namely, working with $H_0 = 74\,{\rm km\,s^{-1}\,Mpc^{-1}}$, see Table 1 and [19], the total (critical) energy density $\rho_{c,0}$ of our spatially flat Universe is at present

$$\rho_{c,0} = \frac{3}{8\pi} M_P^2 H_0^2 = 1.75 \times 10^{-9}\,{\rm eV^4}. \tag{16}$$

The portion of cosmological Dark Matter $\rho_{\rm cdm,0}$ then is, see Table 1,

$$\rho_{\rm cdm,0} \sim 0.35\,\rho_{c,0} \tag{17}$$

which yields a cosmological energy scale $E_{\rm cdm,0}$ in association with Dark Matter of

$$E_{\rm cdm,0} \equiv \rho_{\rm DM,0}^{1/4} \sim 0.00497\,{\rm eV}. \tag{18}$$

On the other hand, we may imagine the percolate of axionic field profiles, which dissolves at $z_p$, to be associated with densely packed dark-matter halos typical of today's galaxies. Namely, scaling

the typical Dark-Matter energy density of the Milky Way $\rho_{\rm dm}$ of Equation (15) from $z_p$ (at $z_p + 0$ $\rho_{\rm dm}$ yet behaves like a cosmological constant) down to $z = 0$ and allowing for a factor of $(\omega_{\rm pdm,0} + \omega_{\rm edm,0})/\omega_{\rm edm,0} = 2.47$, see Table 1, one extracts the energy scale for cosmological Dark Matter $E_{\rm G,0}$ in association with galactic Dark-Matter halos and primordial dark matter as

$$E_{\rm G,0} \equiv \left( \frac{2.47 \rho_{\rm dm}}{(z_p + 1)^3} \right)^{1/4} = (0.00879 \cdots 0.0105)\, {\rm eV}\,. \tag{19}$$

A comparison of Equations (18) and (19) reveals that $E_{\rm cdm,0}$ is smaller but comparable to $E_{\rm G,0}$. This could be due to the neglect of galactic-halo compactification through the missing pull by neighbouring profiles in the axionic percolate and because of the omission of selfgravitation and baryonic-matter accretion/gravitation during the evolution from $z_p = 53$ to present (that is, the use of the values of Equation (15) in Equation (19) overestimates the homogeneous energy density in the percolate). The de-percolation of axionic solitons at $z_p$, whose mean, selfgravitating energy density $\rho_{\rm dm}$ in Dark Matter is nearly independent of cosmological expansion but subject to local gravitation, could therefore be linked to cosmological Dark Matter today within the Dark-Sector model of Equation (9).

## 5. Conclusions

The present article's goal was to address some tensions between local and global cosmology on the basis of the $\Lambda$CDM standard model. Cracks in this model could be identified during the last few years thanks to independent tests resting on precise observational data and their sophisticated analysis. To reconcile these results, a change of $\Lambda$CDM likely is required before the onset of the formation of non-linear, large-scale structure. Here, we have reviewed a proposal made in [34], which assumes thermal photons to be governed by an SU(2) rather than a U(1) gauge principle, and we have discussed the SU(2)$_{\rm CMB}$-implied changes in cosmological parameters and the structure of the Dark Sector. Noticeably, the tensions in $H_0$, the baryonic density, and the redshift for re-ionisation are addressed in favour of local measurements. High-$z$ inputs to CMB and BAO simulations, such as $n_s$ and $\sigma_8$, are sizeably reduced as compared to their fitted values in $\Lambda$CDM. The Dark Sector now invokes a de-percolation of axionic field profiles at a redshift of $z_p \sim 53$. This idea is roughly consistent with typical galactic Dark-Matter halos today, such as the one of the Milky Way, being released from the percolate. Axionic field profiles, in turn, appear to be compatible with Dark-Matter halos in typical galaxies if (confining) Yang–Mills dynamics subject to a much higher mass scales than that of SU(2)$_{\rm CMB}$ is considered to produce the axion mass.

To consolidate such a scenario two immediate fields of investigation suggest themselves: (i) A deep understanding of possible selfgravitating profiles needs to be gained towards their role in actively assisted large-scale structure formation as well as in quasar emergence, strong lensing, cosmic shear, galaxy clustering, and galaxy phenomenology (Tully–Fisher, rotation curves, etc.), distinguishing spirals from ellipticals and satellites from hosts. (ii) More directly, the CMB large-angle anomalies require an addressation in terms of radiative effects in SU(2)$_{\rm CMB}$, playing out at low redshifts, which includes a re-investigation of the CMB dipole.

**Funding:** This research received no external funding.

**Conflicts of Interest:** The author declares no conflict of interest.

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
