# Peer review of "An SU(2) Gauge Principle for the Cosmic Microwave Background: Perspectives on the Dark Sector of the Cosmological Model"

_universe, doi:10.3390/universe6090135_

Round 1

Reviewer 1 Report

I recommend this paper for publication.

Author Response

pls see attachment

Reviewer 2 Report

This paper gives a review of the phenomenological aspects of a cosmological model, based on the assumption that the photons of the CMB are SU(2), instead of U(1), gauge fields. Due to an adjoint Higgs mechanism, below the critical temperature, the thermodynamics of this model leads to a single massless gauge boson (the photon) and two massive vector bosons, whose masses are temperature-dependent. This structure of the theory implies modifications of certain cosmological relations (like, for example, the relation between temperature and redshift) compared to the standard ΛCDM model.

In recent years, there has been an increasing need to modify the ΛCDM cosmological model, in order to resolve the growing tension between the values of the Hubble parameter, determined from local and from global observations. The above-mentioned SU(2) gauge model of the CMB is a novel proposal motivated by this need. It was introduced in Hahn S., Hofmann R., 2017, MNRAS, 469, 1233 [arXiv:1611.02561] and further developed in Hahn S., Hofmann R., Kramer D., SU(2)CMB and the cosmological model: Angular power spectra. MNRAS 2019, 482, 4290.  

The present paper reviews the framework of the above model, as well as the results it leads to regarding the various cosmological parameters. A detailed comparison with the parameters of the standard ΛCDM model is given. The new model is described in an instructive and well-organized manner. The exposition concludes with a discussion of axionic dark matter within this framework. The paper is a well-written and useful review of an interesting recent theoretical development in cosmology. Therefore I recommend its publication.

Author Response

pls see attachment

Reviewer 3 Report

In this work a mechanism already presented in another article is explored in the light of the specific topic of the tension on H0.
The paper is globally well written, with a clear and direct form.
However, some points need to be improved in order to recommend this work for publication:

i) the well-known tension in H0 is quantified in the paper in about 5 \ sigma, which seems an exaggeration. I suggest to indicate the reference or comment more;
ii) I recommend checking the statement : "Recently, a cosmographic way of extracting H0 through time delays of strongly lensed high-redshift 43 quasars, see e.g. [25,26], has reached a precision comparable to that of [19]", explicitly writing the values of H0;
iii) the TT and matter power spectra plot, as also C.L. of parameters, can be presented in the manuscript;
iv) in section 2, the author shows the implications of the model on the universe temperature. Recently, in literature several papers have explored a variation of T0 from that measured by FIRAS, and show that the cosmological data could prefer an higher temperature. How would the evidence of a higher T, with respect to the FIRAS measurements, impact the considerations made in this work?
v) the definition of Neff as "refers to the eff ective number of neutrino flavours" is incorrect. Neff is the effective number of relativistic degrees of freedom at the time of CMB decoupling;
vi) the author claims that the excess of power in low multipoles of TT spectrum could be attributable to the omission of radiative
effects in the low-z propagation of the massless mode. why was this approximation assumed in the analyzes?
vii) comments are required on the compatibility of the found value of omega_b with that estimated by BBN. Show the explicit values with which you are comparing.

Furthermore, I suggest improving some aspects as:
- many acronyms (such as CMB) are never defined, as well as the extended names of the experiments are not reported;
- when defined, the extended name must have the first letter in upper case, see for example how BAO is defined;
- typo in line 38
- several cosmological parameters are also used without being defined first, such as ns and z_re. Provide for an adequate introduction;
- explain the data in detail, for example what Hillipop and TEB are

Author Response

pls see attachment
